# Freezing of Solute-Laden Aqueous Solutions: Kinetics of Crystallization and Heat- and Mass-Transfer-Limited Model

**DOI:** 10.3390/bioengineering9100540

**Published:** 2022-10-10

**Authors:** Stonewall Johnson, Christopher Hall, Sreyashi Das, Ram Devireddy

**Affiliations:** Bioengineering Laboratory, Department of Mechanical Engineering, Louisiana State University, Baton Rouge, LA 70803, USA

**Keywords:** phase change, kinetics of crystallization, differential scanning calorimetry, biological media, latent heat

## Abstract

Following an earlier study, we reexamined the latent heat of fusion during freezing at 5 K/min of twelve different pre-nucleated solute-laden aqueous solutions using a Differential Scanning Calorimeter (DSC) and correlated it with the amount of initially dissolved solids or solutes in the solution. In general, a decrease in DSC-measured heat release (in comparison to that of pure water, 335 mJ/mg) was observed with an increasing fraction of dissolved solids or solutes, as observed in the earlier study. In addition, the kinetics of ice crystallization was also obtained in three representative biological media by performing additional experiments at 1, 5 and 20 K/min. A model of ice crystallization based on the phase diagram of a water–NaCl binary solution and a modified Avrami-like model of kinetics was then developed and fit to the experimental data. Concurrently, a heat and mass transfer model of the freezing of a salt solution in a small container is also presented to account for the effect of the cooling rate as well as the solute concentration on the measured latent of freezing. This diffusion-based model of heat and mass transfer was non-dimensionalized, solved using a numerical scheme and compared with experimental results. The simulation results show that the heat and mass transfer model can predict (± 10%) the experimental results.

## 1. Introduction

The latent heat of fusion in solute-laden aqueous solutions is an important parameter in the modeling and optimization of various low-temperature applications in biomedicine, as well as in the food industry. It is widely reported in the literature that the amount of “freezable” water (or water that changes phase during freezing) is less than the total water content by an amount denoted as the “bound” or “unfreezable” water [1,2,3,4,5,6]. Fennema et al. [7] stated that during the freezing of food substances, the latent heat of fusion should be assumed to be ~80% of the expected heat release based on the total water content. Thus ~20% of the total water content is “bound” and does not freeze in various foods. Further, Cooke and Kuntz [6] reported that as much as 0.8 g of water/g of dissolved solids (to as low as 0.3 g of water/g of dissolved solids) is “bound” or does not freeze in various biological systems (membranes, lipids, intact ribosomes, muscle cells and polypeptides). Thus, there is a need to determine the magnitude of the latent heat of fusion during the freezing of solute-laden aqueous solutions commonly used in cryobiological applications (and, by extension, the amount of “bound” or “non-freezable” water in these biological media) to optimize a variety of freezing applications (including cryopreservation and cryosurgery). 

A DSC is an instrument that measures heat release during a phase change process as a function of time and temperature. This instrument is ideally suited for a variety of measurements of the state of water (particularly liquid-to-solid phase changes) in biological systems and biological media (solute-laden aqueous solutions). A sampling of the contributions in this area during freezing includes the measurement of: (a) the kinetics and extent of ice crystallization in a variety of cryoprotective solutions [8,9,10,11,12,13,14,15]; (b) the “non-freezable” or “bound” water in biological systems [16,17] in solute–macromolecule–water systems [18,19], in food systems [20], in liposomes [21] and in sludge [22]; (c) salt precipitation and latent heat release during the freezing and thawing of phosphate solutions [23]; (d) the effect of antifreeze proteins on the devitrification temperatures of glassy systems [24,25]; and (e) phase and structural transitions in bulk and vicinal water [26]. The DSC has also been used to measure (a) ice nucleation parameters from intracellular ice formation (IIF) heat release readings in yeast and blood cells [27] and in G. Max cells and erythrocytes [28]; (b) heat releases associated with IIF in soybean cells [29], in lymphocytes [30] and in artemia cysts [31]; (c) the volume of freezable water in Drosophila melanogaster embryos [32]; (d) IIF heat release as a function of the cytocrit [33]; and (e) the phase change process in biologically relevant solutions [1,34]. In addition, the DSC has been used to measure the mass transfer (or water transport) across the cell membrane during the freezing of cells in suspension [35,36,37,38,39,40,41,42,43,44,45,46,47,48,49,50,51,52] and cells embedded in tissue [53,54,55,56,57,58,59].

The objective of this study is two-fold: to re-assess the original results from Devireddy et al. [1], which showed that the latent heat of fusion during the freezing of various pre-nucleated aqueous solutions is correlated with the amount of dissolved solids or solutes in the biological media. These measurements were taken using a newer model of a Differential Scanning Calorimeter (DSC-Diamond) as opposed to an older instrument (DSC-Pyris 1) in the earlier study. Similar to the earlier study, the kinetics of ice crystallization were also obtained in three representative biological media (1 × PBS, 10 × PBS and 1M glycerol in 1 × PBS) by performing additional experiments at 1, 5 and 20 K/min. Unlike Devireddy et al. [1], where a full set of heat and mass transfer equations were used to model the process, here, we developed and present a simple model of ice crystallization based on the phase diagram of a water–NaCl binary solution [2] and the modified Avrami-like model of kinetics [3,4,10,60,61]. In the second part of this study, we developed an additional numerical model to describe the heat and mass transfer diffusion problem during the freezing of a salt solution in a small cylindrical container (i.e., a DSC sample pan). This numerical model differs from the one in Devireddy et al. [1] in that no additional experimental parameters are needed to complete the heat and mass transfer model, and the model is self-consistent. Predicted simulation results from the heat and mass transfer model were compared with the corresponding experimental results and show a high degree of agreement. 

## 2. Materials and Methods

### 2.1. Aqueous Solutions: Biological Media

The experiments were conducted using a DSC-Diamond machine (Perkin-Elmer Corporation, Newark, CT, USA). The temperature scale of the instrument was calibrated by the melting point of pure ice (273.15K or 0 °C) and indium (156.7°C for 99.9% purity), while the enthalpy scale was based on the heat of fusion of pure ice (335 mJ/mg), as described earlier [34,35,39]. The latent heat of fusion during freezing was obtained using the DSC in the following solute-laden solutions: (i) 1 × (isotonic), 5 × and 10 × Phosphate-Buffered Saline (PBS) solutions (Celox, Inc., Hopkins, MN, USA); (ii) serum-free RPMI culture media (Celox, Inc., Hopkins, MN, USA); (iii) cell culture media: RPMI with 20% Fetal Bovine Serum (FBS) and 1% penicillin–streptomycin (Sigma Chemical Co., St. Louis, MO, USA); and (iv) 0.05M, 0.1 M, 0.5 M and 1.0 M glycerol in 1 × PBS solutions. Thus, the latent heat of fusion was obtained for nine different aqueous solutions, as described below.

### 2.2. Differential Scanning Calorimeter (DSC) Experiments

The DSC experiments were conducted by placing approximately 9 to 10 mg of each solution in a standard aluminum DSC sample pan (Perkin-Elmer Corporation, Norwalk, CT, USA). The sample was cooled at 5 K/min from 4 °C until ice nucleated in the solution, typically from −6 to −12 °C (observed as a sharp negative peak on the DSC thermogram; Figure 1). The sample was then equilibrated at the phase change temperature (obtained from a separate control experiment as the temperature at which a frozen sample thaws when heated at 5 K/min). The pre-nucleated or “site-saturated” sample was then cooled at 5 K/min to −50 °C to obtain the magnitude and the temperature dependence of the heat release (i.e., the thermogram; see Figure 1). In the cases of 1 × PBS, 5 × PBS, 10 × PBS and 1 M glycerol in 1 × PBS solutions, experiments were also conducted at two additional cooling rates of 1 and 20 K/min. Note that the higher cooling rate of 20 K/min was chosen as a conservative estimate of the fastest cooling rate (40 K/min) at which the DSC can accurately reproduce heat release signatures [35,44,52]. We found that for cooling rates greater than 40 K/min, the DSC heat release measurement spreads out and increases in value [35,44,52]. This inaccuracy could be due to the limitation of the rate at which the phase change process proceeds due to ice crystal growth, as well as the nonlinearity of the resistance within the instrument [62,63,64,65]. Six separate DSC experiments were performed with each solution for each cooling rate studied. The percentage of dissolved solids in the solution was obtained by measuring the difference in weight between hydrated and fully dehydrated solutions (dried in an oven for 3 to 4 d at 50 to 60 °C), the procedure of which was described in an earlier study [1] and, in the interest of brevity, is not repeated here (Table 1).

The integrated area under the DSC thermograms (assumed to correspond to the latent heat of fusion) was obtained using the DSC (Perkin-Elmer Corporation, Norwalk, CT) software with either a sigmoidal or linear baseline, as shown in Figure 1 (as described in the DSC manual and by Devireddy et al. 1998). The choice of the baseline influences the integrated area under the thermogram (i.e., the measured value of latent heat), and although more accurate baseline selections are reported in the literature [62,63,64,65], the simpler sigmoidal and linear baselines were used in this study because of their ease of use and the importance of trends (Table 1). The sigmoidal baseline was drawn between the phase change temperature and ~−22 °C, while the linear baseline was drawn between the phase change temperature and ~−40 °C, as described in the DSC-Diamond manual. 

Note that the sample was equilibrated at the phase change temperature to mimic the behavior of a freezing process in a biological system. For example, water in a cell suspension or a tissue system is compartmentalized into either intracellular or extracellular (vascular in the case of tissues) spaces. During freezing, ice almost always nucleates in the extracellular space (due to the nature of nucleation processes and the fraction of water that is available for phase change). To model the freezing process in such a system, it is necessary to understand the temperature and time dependence of the latent heat released upon subsequent cooling [66]. Hence, the choice was made to pre-nucleate the sample, equilibrate at the phase change temperature and, subsequently, impose a constant cooling rate on the solution.

### 2.3. Temperature (T) and Time (t) Dependence of Latent Heat Release, L (T,t): Avrami-Like Model of Crystallization

As stated earlier, DSC experiments were conducted at three different cooling rates (1, 5 and 20 K/min) for three different solutions (1 × PBS, 10 × PBS and 1 M glycerol in 1 × PBS solution; see Figure 2). Although the total magnitudes of the latent heat release (L) were found to be within 2% of each other for all of the cooling rates studied, there was, however, a time dependence of the measured value of the latent heat release. A function that accounts for the experimentally measured temperature and time dependence of the latent heat release was thus sought. For simplicity, it was assumed that the temperature and time dependence of latent heat release are independent of one another and can be represented as: *L(T,t)=α(T)×β(t)*(1)

This is a very simple assumption that allows for the development of a crystallization kinetics model. Several empirical models were previously fit to the experimentally determined temperature and time dependence of the latent heat release (an example of a purely empirical fit is shown in 64), and the following two functions were chosen because of their possible physical or “mechanistic” significance:(2)$$\alpha \left(T\right)=1-\frac{A}{T_{ph}-T+A}\;\;\;\;\;\;\;\;\;[T<T_{ph}]$$
and
(3)$$\beta (t)=(1-exp(-k\times t^{n}))\;\;\;\;\;\;\;\;\;\;\;[t>0]$$
where “*A*”, “*k*” and “n” are constants that need to be obtained by curve fitting to the experimentally determined data; *T_ph_* is the phase change temperature of the solute-laden aqueous solution (represented as: *T_ph_*= T-osm × 1.858, where *T* is the absolute temperature in degree Kelvin, and Osm is the osmolality of the solution); and *t* is time in seconds and can be represented as: ((*T_ph_*-*T*) × 60)/B, where B is the cooling rate (K/min). 

The function for temperature dependence, *α(T)*, was chosen primarily due to the fact that the temperature dependence of the 1 K/min latent heat release for the 1× PBS solution was found to be very similar to the release profile defined by the phase diagram for an isotonic water–NaCl binary solution, as described previously [2]. Thus, a function that is very similar to the one detailed earlier [2] was also chosen in this study. In fact, if A = 0.53, then Equation (2) corresponds exactly to the function described by Hayes et al. [2] for isotonic water–NaCl solutions. 

The reason for the selection of the time dependence function, *β(t)*, shown in Equation (3), was primarily that it represents the kinetics of transformation, which requires nucleation and/or growth processes, as developed by Avrami [3,60,61] and described in detail by Christian [4]. It should be pointed out that several other models have been reported in the literature that describe crystal growth rates under a variety of conditions in aqueous solutions or otherwise, for example: the studies by Boutron [10], Kubota and Mullin [67] and Hey and MacFarlane [11] and the review by Long et al. [68].
(4)$$L\left(T,t\right)=\left(1-\frac{A}{T_{ph}-T+A}\right)•\left(1-exp\left(-k•t^{n}\right)\right)\;\;\;\;\;\;\;\left[T<T_{ph}\;and\;t>0\;\;\right]$$

The Avrami-like model was chosen over other models in this study because of its simplicity and because, by plotting the data as previously suggested by MacFarlane et al. [69], the DSC-measured heat releases at 5 and 20 K/min were found to follow the Avrami-like kinetics of transformation (Figure 3). The Avrami model of kinetics was originally developed for isothermal crystallization processes and was later extended to non-isothermal processes by Cahn [70,71] and Ozawa [72], with essentially the same formulation as shown above. The exception being that for non-isothermal processes (similar to the experiments performed in the present study), the constant, k, is a function of temperature, T. To simplify the curve-fitting process and the Avrami-like model, this dependence of k on T was neglected in the present study. Clearly, this assumption represents a first-order approximation of the traditional Avrami models of crystallization. The two constants in the current Avrami-like model, “*k*” and “*n*”, are a constant and the time exponent, respectively. The time exponent, *n*, is characteristic of the nucleation type and the growth geometry, and the constant, *k*, is related to the nucleation and growth rates [4]. 

Briefly, in the case of diffusion-controlled growth, the time exponent, n, is expected to be 1.5 for “site-saturation” or for all nuclei present at the beginning of the transformation with negligible initial dimensions or 2.5 when the nucleation rate is constant during the transformation process. When the growth takes place in large plates or cylinders, the exponent is reduced to 0.5 and 1, respectively. For polymorphic changes, discontinuous precipitation, eutectoid reactions and interface-controlled growth, the time exponent, *n*, can vary between 1 and 4, depending on the conditions of growth (for “site-saturation” or “a nucleation process in which all nucleation sites are exhausted early in the transformation”, it varies between 1 and 3; for a constant nucleation rate, it takes a value of 4, and it has a value between 3 and 4 for a decreasing nucleation rate). The constant, *k*, is a measure of the crystal growth rate, and a higher value of k signifies a faster rate of crystal growth or transformation. Note that the predicted value for the constant k in Table 2 ranges from 3.3 to 2.3, with the presence of glycerol decreasing the value, suggesting that the presence of glycerol saturates the nucleation sites when compared to plain PBS solutions. 

The constant “*A*” in the temperature dependence function, *L(T)*, was obtained using the 1 K/min data for the three different solutions investigated (1 × PBS, 10 × PBS and 1 M glycerol in 1 × PBS) by using a least-square minimization technique [73]. After determining the best-fit value of “*A*” to the 1 K/min data, the function *L(T,t)* was fit to the 5 and 20 K/min data, and the two remaining unknowns in the model (constant, *k*, and time exponent, *n*) were obtained for the three different solutions studied. In addition, the “combined best-fit” values of the constant, *k*, and time exponent, *n*, that best fit to the 5 and 20 K/min data concurrently were also obtained (*a* similar procedure was previously applied for other functions [36,44,54,74]). All of the curve-fitting results presented have an R^2^ value greater than or equal to 0.95, indicating that there was good agreement between the experimental data points and the fit calculated using the estimated constants.

### 2.4. Heat- and Mass-Transfer-Limited Model of Freezing of a Salt Solution in a Small Container

We start by considering the problem of freezing a salt solution in a small container (or a DSC sample pan) as a coupled problem of the temperature and salt concentration. Let H be the height of the salt solution, λ be the mean thickness of dendritic fingers (assumed to be 5 × 10^−5^ m) and R be the radius of the cylinder (2 × 10^−3^ m). The volume of the fluid is: $V=\pi R^{2}H$. Since the pan is heated from below (axial direction), we can try to model the temperature T and the concentration c as having radial symmetry, i.e., $T\left(r,z,t\right)=T\left(z,t\right),$ and $c\left(r,z,t\right)=c\left(z,t\right).$ Let $Z\left(t\right)$ be the position of the freezing front at time t. Then, the governing dimensional equations are: (5)$$\frac{\partial T}{\partial t}\left(z,t\right)=D_{T,i}\frac{\partial^{2}T}{\partial z^{2}},\;\;\mathrm{for}\;i=1,\;2$$
(6)$$\frac{\partial c}{\partial t}\left(z,t\right)=D_{c,i}\frac{\partial^{2}c}{\partial z^{2}},\;\;\mathrm{for}\;i=1,\;2$$
where i = 1 denotes the frozen region $0\leq z\leq Z\left(t\right)$, and i = 2 denotes the unfrozen region $Z\left(t\right)\leq z\leq H$. The boundary conditions at the free boundary $z=Z\left(t\right)$ are: (7)$$T\left(Z^{+}\left(t\right),t\right)=T\left(Z^{-}\left(t\right),t\right)=T_{ph}=273.15-mc{\left(Z^{+}\left(t\right),t\right)}_{,}$$
(8)$$L\dot{Z}\left(t\right)=k_{1}\frac{\partial T}{\partial z}\left(Z^{-}\left(t\right),t\right)-k_{2}\frac{\partial T}{\partial z}{\left(Z^{+}\left(t\right),t\right)}_{,}$$
(9)$$\dot{Z}\left(t\right)(c(Z^{-}\left(t\right),t)-c\left(Z^{+}\left(t\right),t)\right)=D_{c,2}\frac{\partial c}{\partial z}\left(Z^{+}\left(t\right),t\right)-D_{c,1}\frac{\partial c}{\partial z}\left(Z^{-}\left(t\right),t\right)$$
where $T\left(Z^{+}\left(t\right),t\right)=lim_{z\to Z^{\pm }\left(t\right)}T\left(z,t\right)$. The initial conditions and the boundary conditions at the ends z=0 and z=H. are: $T\left(0,t\right)=T_{ph}-Bt,$ the temperature decreases linearly from the phase change temperature (the imposed constant cooling rate $c\left(0,t\right)=0$), the frozen fraction is pure ice, $\frac{\partial T}{\partial z}\left(H,t\right)=0$ and $\frac{\partial c}{\partial z}\left(H,t\right)=0$, Neumann conditions. The initial conditions are: $T\left(z,0\right)=T_{ph}=273.15-mc_{0}$, and $c\left(z,0\right)=c_{0}$. Note that m is a constant that defines the phase change temperature based on the initial solute concentration,$\;c_{0}$, and is approximately equal to 1.858 (if $c_{0}$ is given in Osm/L; for an isosmotic solution corresponding to a NaCl concentration of 0.9 wt%, the $c_{0}$ value is 0.3 Osm/L). Assuming that the diffusion in the frozen region is negligible ($D_{c,1}=0\;$and $c\left(z,t\right)=0$ for $0\leq z\leq Z\left(t\right)$) so that the frozen fraction is pure ice, we can non-dimensionalize the equations using the length scale l=H and the time scale τ=TphB with the following non-dimensional quantities: T´=TTph, c´=cc0, z´=zl and t´=tτ. Dropping the primes and simplifying, we obtain the following non-dimensional system: $\varepsilon_{i}\frac{\partial T}{\partial t}=\frac{\partial^{2}T}{\partial z^{2}}$ and $\gamma_{2}\frac{\partial c}{\partial t}=\frac{\partial^{2}c}{\partial z^{2}}$ for $Z\left(t\right)\leq z\leq 1$, where $\varepsilon_{i}=\frac{BH^{2}}{T_{ph}D_{T,i}}$ and $\gamma_{2}=\frac{BH^{2}}{T_{ph}D_{c,2}}$ are the inverse thermal and compositional diffusivities, respectively. The boundary conditions are given by: (10)$$T\left(Z^{\pm }\left(t\right),t\right)=1-M(c(Z^{+}(t)-1),\;\;\;\;\;\;\;\;\;\;\;\;St\varepsilon_{2}\dot{Z}=k\frac{\partial T}{\partial z}\left(Z^{-}\left(t\right),t\right)-\frac{\partial T}{\partial z}\left(Z^{+}\left(t\right),t\right)$$
(11)$$\gamma_{2}\dot{Z}=\frac{-1}{c\left(Z^{\pm }\left(t\right),t\right)}\;\frac{\partial c}{\partial z}\left(Z^{+}\left(t\right),t\right),\;\;\;\;\;\;\;\;\;\;\;\;T\left(0,t\right)=1-t,\;\;\;\;\;\;\;\;\;\;\;\frac{\partial T}{\partial z}\left(1,t\right)-\frac{\partial c}{\partial z}\left(1,t\right)=0\;\;$$
where ***St*** is the Stefan number ($\frac{L}{T_{ph}c_{l}}\;\sim 10^{-2}$), M = $\frac{mc_{0}}{T_{ph}}$, k = $\frac{k_{1}}{k_{2}}$, and $c_{l}$ is the heat capacitance of the liquid. The initial conditions are: $T\left(z,0\right)=1$, and $c\left(z,0\right)=1.$ We now estimate the non-dimensional coefficients: $\varepsilon_{i}=\frac{BH^{2}}{T_{ph}D_{T,i}}\;\sim B•2\times 10^{-4}$ min/K, assuming H = $5\times 10^{-5}\;to\;5\times 10^{-4}\;m$, and DT,i~10−5 to 10−7 m2s. Similarly, $\gamma_{2}=\frac{BH^{2}}{T_{ph}D_{c,2}}\;\sim B•2\times 10^{-1}$ min/K, assuming $D_{c,2}\sim$ 10−9 m2s. Note that $\varepsilon_{i}\;$is quite small, even for moderate cooling rates, while $\gamma_{2}$ is only negligible for very low cooling rates, and that $St\varepsilon_{i}\;\ll \;\varepsilon_{i}$. This suggests that we can consider a further ***reduced system*** such that the heat conduction is quasi-steady, while the full diffusion equation is used for the solute concentration. That is, $\frac{\partial^{2}T}{\partial z^{2}}=0$ for i = 1, 2, and $\gamma_{2}\frac{\partial c}{\partial t}=\frac{\partial^{2}c}{\partial z^{2}}$ for $Z\left(t\right)\leq z\leq 1$. The boundary conditions for the reduced system are:(12)$$T\left(Z^{\pm }\left(t\right),t\right)=T_{ph}=1-M(c(Z^{+}(t)-1),\;\;\;\;\;\;\;\;\;\;\;\;0=k\frac{\partial T}{\partial z}\left(Z^{-}\left(t\right),t\right)-\frac{\partial T}{\partial z}\left(Z^{+}\left(t\right),t\right)$$
(13)$$\gamma_{2}\dot{Z}=\frac{-1}{c\left(Z^{\pm }\left(t\right),t\right)}\;\frac{\partial c}{\partial z}\left(Z^{+}\left(t\right),t\right),\;\;\;\;\;\;\;\;\;\;\;\;T\left(0,t\right)=1-t,\;\;\;\;\;\;\;\;\;\;\;\frac{\partial T}{\partial z}\left(1,t\right)-\frac{\partial c}{\partial z}\left(1,t\right)=0.\;\;\;\;\;\;\;\;\;\;\;\;\;\;\;\;\;\;\;\;$$

The initial data are $T\left(z,0\right)=c\left(z,0\right)=1$. The solution to the temperature field is trivial: $T\left(z,t\right)=1-t$. Thus, from the first boundary condition of the reduced system, we find that $c\left(Z^{+}\left(t\right),t\right)=1+\frac{t}{M}$. Finally, the system reduces to only solving the diffusion equation for c in the liquid fraction: (14)$$\gamma_{2}\frac{\partial c}{\partial t}=\frac{\partial^{2}c}{\partial z^{2}}\;\mathrm{for}\;Z\left(t\right)\leq z\leq 1$$
with γ2Z˙=−11+tM ∂c∂zZ+t,t,            ∂c∂z1,t=0,            and cZ+t,t=1+tM.

This is the model that was solved numerically by reformulating the equation in a fixed domain. Rather than solving the diffusion Equation (14) on the variable domain $Z\left(t\right)\leq z\leq 1$, we mapped the problem onto a fixed domain: 0≤δ≤1. This is carried out as follows: First, we define the parametrization $z=z\left(\delta ,t\right)\;$such that $z\left(\delta ,t\right)$ is uniform in δ. We further suppose that $z\left(0,t\right)=Z\left(t\right)$ is the interface position. In addition, $z\left(1,t\right)=1$; it is now easy to see that: $z\left(\delta ,t\right)=(1-Z\left(t\right)\delta +Z\left(t\right).$ Transforming Equation (14), we obtain: (15)$$\frac{\partial c}{\partial z}\left(\delta ,t\right)=-\frac{\dot{Z}\left(t\right)\left(\delta -1\right)}{1-Z\left(t\right)}\;\frac{\partial c}{\partial \delta }+\frac{1}{\gamma_{2}}\;\frac{1}{{\left(1-Z\left(t\right)\right)}^{2}}\;\frac{\partial^{2}c}{\partial \delta^{2}}$$
(16) γ2Z˙=−11+tM 11−Zt ∂c∂z0,t,            c0,t=1+tM,            ∂c∂δ1,t=0, cδ,0=1

Equation (15) was solved with boundary conditions (Equation (16)) using a first order in time and a second order in space. 

### 2.5. Numerical Scheme

Let h=1N+1 be the spatial grid size, and let the grid be given by $\delta_{j}=jh$, with $\delta_{0}=0$ and $\delta_{N+1}=1$. Let $c_{j}\left(t\right)=c\left(\delta_{j},t\right)$. Then, the semi-discrete scheme is: (17)$$\frac{d}{\partial t}c_{j}\left(t\right)=-\frac{\dot{Z}\left(t\right)\left(\delta_{j}-1\right)}{1-Z\left(t\right)}\;\frac{c_{j+1}\left(t\right)-c_{j-1}\left(t\right)}{2h}+\frac{1}{\gamma_{2}\left(t\right)}\;\frac{1}{{\left(1-Z\left(t\right)\right)}^{2}}\;\frac{c_{j+1}\left(t\right)-2c_{j}\left(t\right)+c_{j-1}\left(t\right)}{2h}$$
(18) with γ2tZt˙=−11+tM 11−Zt c−1t−c+1t2h with j=1,………. N+1

The initial and boundary conditions are: cj0=1,cN+2t=cN−1t, c0t=1+tM. For the boundary value, $\;c_{-1}=3c_{0}-3c_{1}+c_{2}$. For the inverse non-dimensional coefficient $\;\gamma_{2}\left(t\right)$, we begin by considering the dimensional diffusion coefficient $D_{c}\left(t\right)=\frac{k}{6\pi R}•\frac{T}{\mu }$, with D (298K) = 2.83 × 10^−11^ m^2^/min and viscosity (μ) = 1cP, to obtain $\frac{k}{6\pi R}=9.17\times 10^{-14}$. For the viscosity model, we assume the well-known formulation: μ=AeFRT, with A = 6.627 × 10^−4^, F = 1.807 × 10^4^ and R = 8.314, to calculate the non-dimensional inverse diffusion coefficient, $\;\gamma_{2}=\frac{BH^{2}}{T_{ph}D_{c}\left(T\right)}\;$. Several studies have shown that there is a strong influence of solute concentration on solution viscosity [75,76,77,78,79,80]. This effect was not considered in the present study to reduce the complexity of the model and also to assess whether a simpler approach would suffice. For the fully discrete scheme (in time), we let Δt be the time step and let $t_{n}=n\Delta t$. The scheme is then given by $Z^{n+1}=Z^{n}+\Delta t•{\dot{Z}}^{n}$, where Z˙n=−1γ2tn•11+ tnM•11−Zn•c−1t−c+1t2h. The concentration profile then evolves to: $c_{j}^{n+1}=c_{j}^{n}-\Delta t\left[\frac{\dot{Z^{n}}\left(\delta_{j}-1\right)}{1-Z^{n}}•\frac{c_{j+1}\left(t\right)-c_{j-1}\left(t\right)}{2h}-\frac{1}{\gamma_{2}\left(t_{n}+1\right)}•\frac{c_{j+1}^{n+1}-2c_{j}^{n+1}+c_{j-1}^{n+1}}{h^{2}}\right]$. for j=1,………. N+1. The initial and boundary conditions are: cj0=1,cN+2n=cN−1n, c−1=3c0−3c1+c2, c0n=1+tnM.

## 3. Results

As stated earlier, we tried to reproduce the earlier results of the magnitude of latent heat during freezing obtained by Devireddy et al. [1]. Table 1 shows the DSC-measured heat release readings for the various aqueous solutions investigated. In general, these results are within ±1% of the results obtained by Devireddy et al. [1] and show a decrease in heat release as the amount of dissolved solids (solutes) increases. This trend/result is consistent with earlier studies [1,34]. Note that, as expected, the dissolved fraction of solids increases with the solute concentration, i.e., from 1 × PBS to 10 × PBS. As demonstrated in the earlier study [1] and reconfirmed in this study, although serum-free RPMI has approximately the same fraction of dissolved solids as the cell culture media, it has a considerably smaller magnitude of latent heat released (261 vs. 221 mJ/mg). As stated in earlier studies [18,19,75], this decrease in latent heat release might either be due to the decrease in the latent heat of water (at lower temperatures) and/or due to the presence of “bound” water. 

Figure 2A–C show the experimentally determined temperature dependence of latent heat release from 1 × PBS, 10 × PBS and 1M glycerol in 1 × PBS solutions, respectively. In each figure, the experimentally determined fraction of heat release at various sub-zero temperatures is shown: 1 °C/min (filled circles), 5 °C/min (open squares) and 20 °C/min (filled triangles). Note that the heat release profile of the 1 × PBS solution for a cooling rate of 1 °C/min is very similar to the release profile defined by the phase diagram for a water–NaCl binary solution (solid line in Figure 2A). Figure 2 also shows that the heat release profiles for 5 and 20 °C/min significantly lag behind the 1 °C/min profile for all three solutions studied. The constant “A” obtained by curve fitting to the 1 °C/min data was found to be 0.53 for all three solutions investigated (Table 2). Thus, the temperature dependence function shown in Equation (2), when applied to the 1 × PBS solution, is exactly the same as the one described previously by Hayes et al. [2] for a binary (water–NaCl) isotonic solution. The solid lines in Figure 2B,C represent the temperature dependence function shown in Equation (2), with A = 0.53 for 10 × PBS (*T_ph_* = ~−5.3 °C or ~267.85 K) and for 1 M glycerol in 1 × PBS (*T_ph_* = ~−2.4 °C or ~270.75 K), and are shown to accurately predict the fraction of heat release measured at 1 °C/min.

Figure 3 shows the plots of the 1, 5 and 20 °C/min data for the 1 × PBS solution, as previously suggested by MacFarlane et al. [69]. Equation (3) can be recast in the form: *ln*[-ln(1-β(t))]=ln(κ)+γ•ln(t), and thus, plots of *ln*[-ln(1-β(t))] against ln(*t*) should fall on a straight line when the transformation process follows the model, with the slope of the line being equal to the time exponent “n” and the y-intercept being equal to the natural log of the constant, k, or ln(k). Such a plot is shown in Figure 3 and shows that the 1 °C/min (filled circles), 5 °C/min (open squares) and 20 °C/min (open triangles) data points can be represented by three separate straight lines. Thus, Figure 3 suggests that the transformation process follows the Avrami-like model reasonably well and validates our choice of L(t) shown in Equation (3). The three straight lines from left to right represent lines with a slope of 1.5 (or the time exponent, n = 1.5). The 1 °C/min (filled circles), 5 °C/min (filled squares) and 20 °C/min (open triangles) data points are found to be reasonably well represented, even with the solid lines. The time exponent, n, was then set to 1.5 for two reasons: (1) it corresponds to the physical situation of the DSC experiment: i.e., at time t = 0, all nuclei are present or “site-saturated” [4], and (2) as is shown later (Figure 4), the value of “n” that best fits the 1, 5 and 20 °C/min data concurrently was also found to be equal to 1.5 (the “combined best-fit” value of “n” was also found to be 1.5 for the other two solutions studied, as shown in Table 2 and Figure 4).

Figure 4A–C show contour plots of the goodness-of-fit parameter, *R^2^*, in the constant (k) and time exponent (n) space that “fit” the 5 and 20 °C/min data for 1 × PBS, 10 × PBS and 1 M glycerol in 1 × PBS solutions, respectively. Any combination of “k” and “n” shown to be within the contour will “fit” the experimentally determined water transport data at that cooling rate with an *R^2^* value > 0.95. The common region between the contours indicates the combination of “k” and “n” that will fit the data at both 5 and 20 K/min with an *R^2^* ≥ 0.95. The predicted “combined best-fit” values of “k” and “n” are denoted by a star (*) and fall within the two contours. The “combined best-fit” values of the constant, k, and the time exponent, n, are shown in Table 2, along with the “best-fit” values of the constant “A”, for the three different solutions investigated in this study (1 × PBS, 10 × PBS and 1 M glycerol in 1 × PBS solutions).

**Heat- and Mass-Transfer-Limited Model Results:** In Figure 5, we present comparisons between the numerical and experimental results for the percentage of heat released as a function of -(*T*-*T_ph_*) obtained using the mathematical model described earlier, Equations (10) and (11). Given that the magnitude of the heat of fusion cannot be easily calculated from the heat and mass transfer model, the percentage of heat released was calculated directly from the interface position by assuming t*_final_* for each simulation as the time at which *T*(t*_final_* )=*T_ph_*-40. Then, the numerical percentage of latent heat released is: *% Latent Heat released (t) = (Z(t))/(Z(t_final_))*. This is consistent with the experimental value if the latent heat produced by freezing an infinitesimal amount of ice does not vary with temperature/time. 

In Figure 5, there are three plots. In each plot, a different initial concentration of salt/PBS is used; Figure 5A represents 1 × PBS, and 5B shows 5 × PBS, while 5C shows the 10 × PBS solution. In each graph, the results are shown using cooling rates of 1, 5 and 20 K/min, as well as the corresponding experimental results. Additionally, the solid curve in each plot corresponds to the % of latent heat released that is predicted by using the solution in the phase diagram, which assumes that the concentration is equilibrated (constant). Observe that as the cooling rate is decreased, the numerical solution converges to that predicted by the phase diagram. Moreover, the % of latent heat released is a decreasing function of the cooling rate in all cases investigated. There is good agreement between the numerical and experimental results with 1 ×, 5 × and 10 × PBS. It is intriguing to note that the shape of the experimental curves is insensitive to the initial concentration. The experimental curves seem to be merely shifts of one another, with the shift reflecting the lowering of the phase change temperature with the increase in the initial concentration. A slightly different behavior is seen in the numerical solution, with the simulations at higher cooling rates under-predicting (~ 6 to 10%) the amount of latent heat released. Given the simplicity of the assumed numerical model, this error was deemed to be acceptable. However, increased agreement between the heat and mass transfer limited model simulations and the experimental results could possibly be achieved by varying/optimizing the mean thickness of the dendritic fingers between the 1 ×, 5 × and 10 × PBS solutions, as described in [1]. Additional improvements to the model are presented in the discussion section.

## 4. Discussion

The latent heat data presented in this study suggest that the previously described recommendation by Fennema et al. [7] that ~80% of total water content freezes in food substances results in an over-prediction of the latent heat of fusion of aqueous solutions (by as much as ~45% for 5 × PBS) and also that the suggested value of bound water (~0.3 to 0.8 g of water/g of dissolved solids in membranes, lipids, intact ribosomes, muscle cells and polypeptides) by Cooke and Kuntz [6] is a very conservative estimate (~10 times lower for 1 × PBS). An important cryosurgical application of the lower value of the latent heat of fusion of aqueous solutions (vs. in pure water) is to increase the size of ice balls formed during the freezing of a solute-laden aqueous solution, as compared to that in pure water, for a specified cooling load [81].

The magnitude of the latent heat of freezing reported in the current study is in good agreement with a similar study performed in 2002 by Devireddy et al. [1] and is also in agreement with the previously published literature. For example, values of 275 to 250 mJ/mg were reported for phosphate and sodium buffer solutions by Murase and Franks [23] and were obtained using a Differential Scanning Calorimeter (DSC-2). Similarly, values ranging from 29.3 to 218 mJ/mg were reported by Iijima [12] when solutions containing glycerol (60 to 10% wt/v ratio) were thawed at 10 K/min in a DSC-7. Similar results were obtained by Han and Bischof [34] utilizing a DSC-Pyris 1 machine. Other investigators have reported similar trends [82,83,84,85,86,87,88,89,90]. Several theories have been proposed to explain the inverse relationship between the solute concentration and measured latent heat values, including temperature effects, unfreezable or bound water, the solute distribution and the associated heats of dissolution and entropic effects, and are not repeated here. The interested reader is referred to the primary sources and review articles [5,6,7,75,84,85]. 

A function based on the phase diagram of a binary solution and the modified Avrami-like model of kinetics was developed to predict the measured temperature and time dependence of the latent heat release. The modified Avrami model of kinetics, as a first-order approximation, disregards the temperature dependence of *k;* in the traditional model of Avrami kinetics, this represents a process that is isothermal. An additional simplifying assumption is in Equation (1), where it is assumed that the temperature and time dependence of the latent heat release are independent of each other and that the combined response can be modeled as a superposition of the two effects. The validity of these assumptions is supported by the ability of the model to predict the experimental results, but clearly, this does not suggest that the simplified Avrami-like model presented here fully captures the underlying mechanisms/physics of the freezing process. However, the presented model is quite simple and straightforward to apply in numerical schemes of biological freezing rather than incorporating the full equations of heat and mass transfer; see, for example, Devireddy et al. [66] for an example of such a coupled freezing problem in tissues. The model as developed is applicable to PBS-based solutions as well as PBS–glycerol solutions. In theory, this model should be applicable to all solute-laden solutions. However, additional experimental data are needed to verify this assumption/claim.

Three different model constants were obtained by curve fitting to the experimental data to complete the function (Equation (1)). The constant in the temperature dependence function, A, was found to be 0.53 from the 1 K/min data for the three different solutions studied. Significantly, the “combined best-fit” value of the time exponent, n, also remained constant at 1.5 for all three solutions studied (Table 2 and Figure 5) and is presumably due to the fact that, in the model, n = 1.5 correlates with the physical situation of the DSC experiment, i.e., pre-nucleated solutions at time t = 0 or “site-saturated” conditions [4]. Another interesting observation is that the value of “k” remains constant between 1 × PBS and 10 × PBS solutions at 3.3 and falls to 2.3 for 1 M glycerol in 1 × PBS solution (Table 2). As mentioned earlier, the constant, k, can be thought of as a measure of the crystal growth rate or the rate of the transformation process, and a lower value of “k” implies a slower rate of growth. Thus, the reduction in the value of “k” between the PBS (1 × and 10 ×) and glycerol in 1 × PBS solutions suggests that the crystal growth rate or the rate of transformation is slower in the latter (glycerol solution) in comparison with the former solutions (1 × and 10 × PBS). The slower crystal growth rate in the glycerol solution is presumably due to its higher viscosity in comparison to the PBS solutions; i.e., the “type” of solute affects the value of “k” and not the “amount” (since “k” is constant between 1 × and 10 × PBS solutions). 

A closer examination of Figure 2 shows that the fit of Equation (1) to the data shows a cooling rate dependence. This suggests a model limitation that is clearly due to the use of an isothermal model for a non-equilibrium process. Specifically, at the fastest cooling rate studied, 20 K/min, Equation (1) under-predicts the fraction of heat released for 1 × PBS (*T_ph_* < *T* < −15 °C), while the opposite is true for both the 10x PBS (−12 < *T* < −22 °C) and 1M glycerol in 1 × PBS (*T* < −12 °C) solutions. No satisfactory explanation is available at this time for this observation, apart from the ones stated earlier. Note that a similar trend of under-prediction is seen in the numerical results as well (Figure 5). It might be that both the Avrami-like model and the numerical model are missing a fundamental piece of the puzzle, or the experimental data are showing an artifact of unknown origin. For example, the choice of the model diffusion coefficient (or the viscosity) could be incorrect. However, a sensitivity analysis found that the model predictions will “match” the experimental data only if the diffusion coefficient is lowered by a factor of 10^−10^. This decrease in the diffusion coefficient is neither unsupported by experiments nor realistic. Effects such as the temperature dependence of solute diffusivity and the temperature dependence of latent heat were considered but were not significant enough to change the model results, unless unrealistic assumptions that are unsupported in literature were made, in a similar fashion to the diffusion coefficient. Additional modifications and model improvements were considered but were deemed to be too impractical, as they required further assumptions with several unknown variables and also increased the complexity of the model to be developed. For example, model improvements include the inclusion of ice crystal interactions, the inclusion of nucleation models to assess the ice crystal size and distribution, the inclusion of irregularly shaped and sized ice crystals, the modeling of the formation of partial and/or complete eutectics at the advancing ice front, the inclusion of salts in the frozen fraction and instabilities at the advancing ice front. 

## 5. Conclusions

The latent heat of fusion during the freezing of different pre-nucleated solute-laden aqueous solutions was obtained using a Differential Scanning Calorimeter (DSC) and correlated with the amount of initially dissolved solids or solutes in the solution. In general, a decrease in DSC-measured heat release (in comparison to that of pure water, 335 mJ/mg) was observed with an increasing fraction of dissolved solids in the solution, a fact that has been well established in the published literature. A model based on the phase change diagram of a water–NaCl binary solution and a modified Avrami-like model of kinetics was developed and fit to the observed data to obtain three model parameters (“A”, “k” and “n”). The model was found to simulate the temperature and time dependence in the DSC-measured heat release data reasonably well (the goodness-of-fit parameter *R^2^* ≥ 0.95). A mathematical model of the freezing of a salt solution in a small container is also presented to further describe the experimental measurements. This model was non-dimensionalized, solved using a numerical scheme and compared with experimental results. The results show that the simulations of the mathematical model are in good agreement (± 10%) with the experimental results. 

## Figures and Tables

**Figure 1 bioengineering-09-00540-f001:**
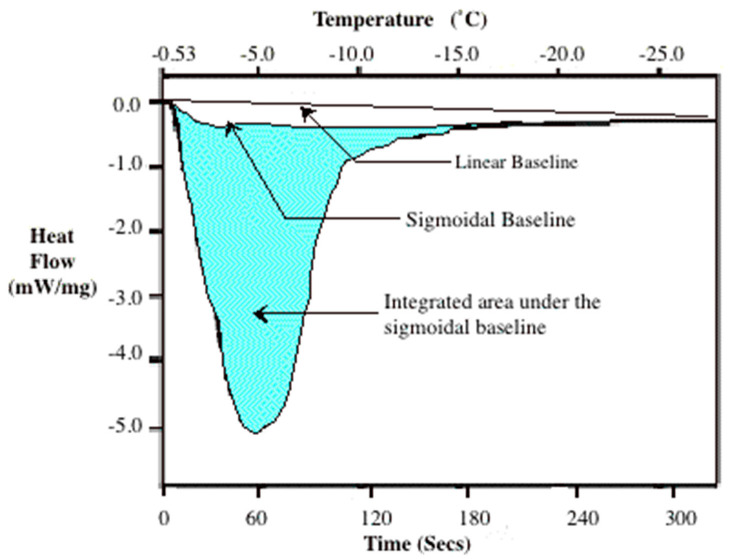
A typical DSC thermogram showing the latent heat release (area under the curve) using a linear and sigmoidal baseline. The integrated area (shaded region) under the sigmoidal baseline represents the magnitude of the latent heat released during freezing. Note that a negative value on the y-axis represents an exothermic reaction. Redrawn with permission from Devireddy et al. [1].

**Figure 2 bioengineering-09-00540-f002:**
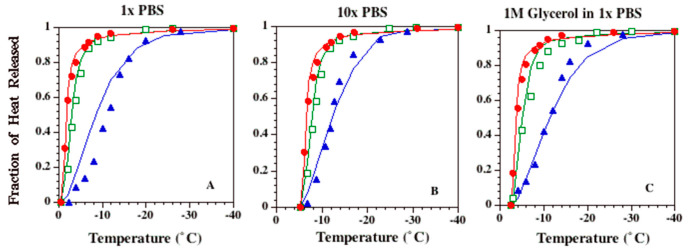
Temperature and time dependence of the DSC-measured heat release. Figure 2A–C show the experimentally determined temperature dependence of latent heat release from 1 × PBS, 10 × PBS and 1 M glycerol in 1 × PBS solutions, respectively. In each figure, the experimentally determined fraction of heat release is shown on the y-axis, and the sub-zero temperatures are plotted on the x-axis. In all of the figures, the filled circles represent the 1 K/min data, the open squares represent the 5 K/min data, and the filled triangles represent the 20 K/min data. Avrami-like model simulations using Equations (1)–(3), along with the best-fit parameters shown in Table 2, are also shown in the figures. Error bars are present but are too small to resolve in the graph (n = 6 DSC cooling runs at each cooling rate for each solution).

**Figure 3 bioengineering-09-00540-f003:**
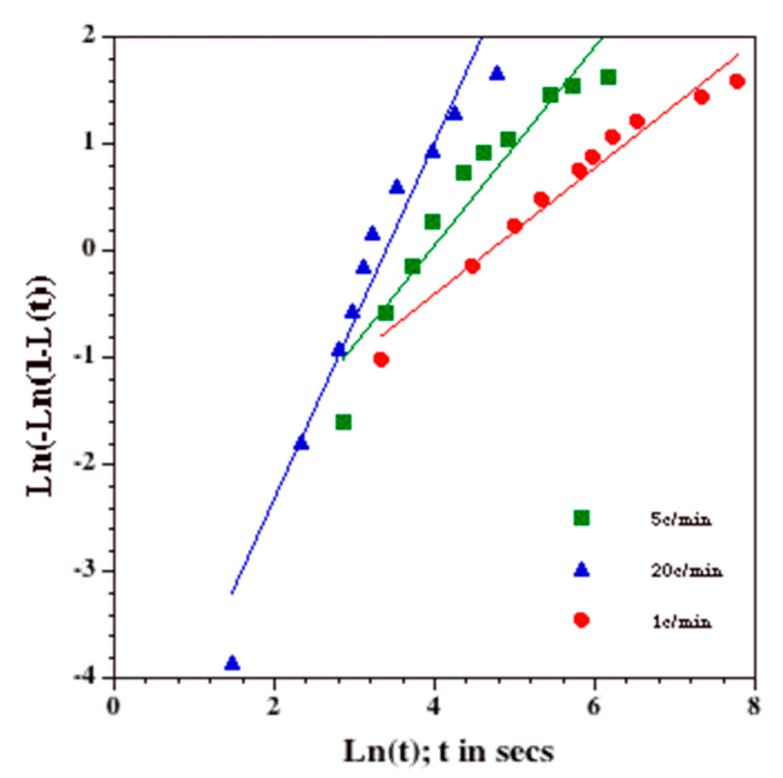
Linearized Plot of Equation (3): The 1 K/min (filled circles), 5 K/min (filled squares) and 20 K/min (filled triangles) data for 1 × PBS solution plotted using a linearized version of the Avrami-like kinetic model. ln[−ln(1 − β(t))] is plotted on the y-axis, while ln(t) is plotted on the x-axis (time, t, is in secs). The solid straight lines represent lines with a slope of 1.5 (or the time exponent, n = 1.5).

**Figure 4 bioengineering-09-00540-f004:**
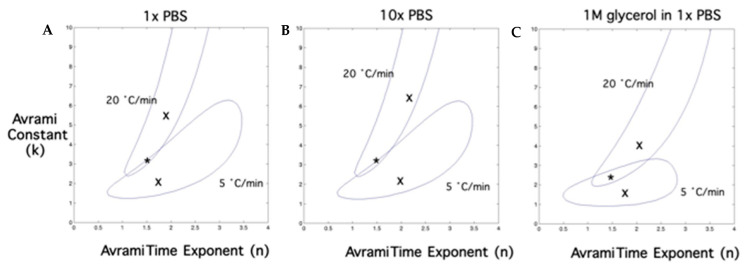
Contour plots of the goodness-of-fit parameter R^2^ (≥ 0.95) for the Avrami-like kinetic model. Figure 4**A**–**C** show the combinations of constant (k) and the time exponent (n) that fit the 5 and 20 K/min data for 1 × PBS, 10 × PBS and 1M glycerol in 1 × PBS solutions, respectively, with an R2 ≥ 0.95. Note that the contour plot for 1 K/min is completely enclosed by the contour plot corresponding to 5 K/min for all media (contour not shown in the interest of clarity). The “combined best-fit” parameters for 1, 5 and 20 K/min are represented by a star (*) in each figure and are: n = 1.5 and k = 3.3 for 1 × and 10 × PBS solutions and n = 1.5 and k = 2.3 for 1M glycerol in 1 × PBS solution (Table 2). The constant (k) is plotted on the y-axis, while the time exponent (n) is plotted on the x-axis.

**Figure 5 bioengineering-09-00540-f005:**
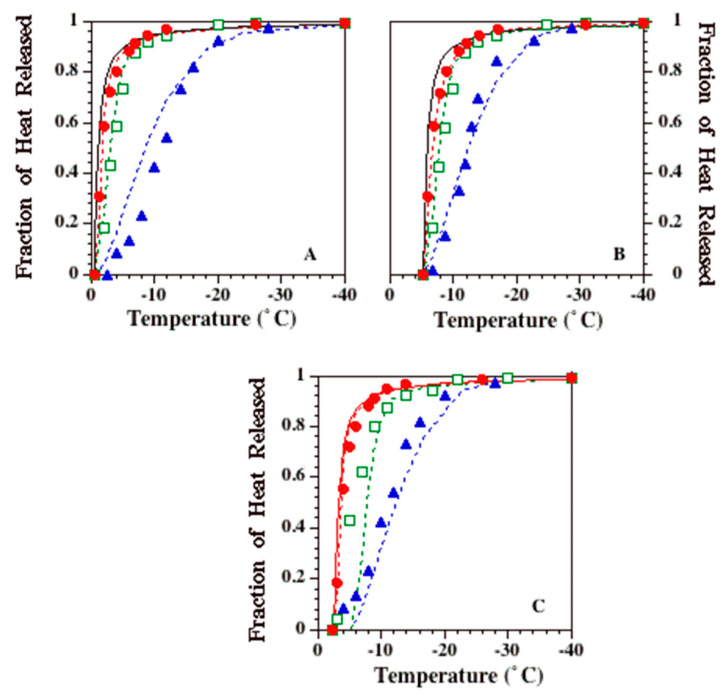
Model simulations of L(T,t) using Equations (15) and (16). The three figures represent the three different concentrations of PBS (1 ×, 5 × and 10 ×). Each graph contains simulated (heat- and mass-transfer-diffusion-limited model) and experimental results from equilibrium cooling (phase diagram solution, solid lines), 1 K/min (filled circles), 5 °C/min (open squares) and 20 K/min (filled triangles). In each figure, the three curves (dotted lines), from left to right, represent the model-simulated fraction of heat release at 1, 5 and 20 K/min, respectively. In each figure, the experimentally determined fraction of heat release is shown on the y-axis, and sub-zero temperatures are plotted on the x-axis.

**Table 1 bioengineering-09-00540-t001:** DSC-Measured Heat Release as a Function of % Dissolved Solids in Solute-Laden Aqueous Solutions (Biological Media).

Aqueous Solution	Magnitude of DSC-Measured Heat Release (J/g)		% of Dissolved Solids
	Linear Baseline (Tph to ~–40 °C)	Sigmoidal Baseline (Tph to ~–22 °C)	
Pure Water	335.0 ± 5.0	335.0 ± 5.0	0.0
*PBS:*			
1 × PBS	302.0 ± 5.0	256.0 ± 5.0	0.98 ± 0.05
5 × PBS	221.0 ± 5.0	183.0 ± 5.0	5.34 ± 0.07
10 × PBS	174.0 ± 5.0	142.0 ± 5.0	9.81 ± 0.04
*Glycerol in 1 × PBS*:			
0.05 moles	294.0 ± 5.0	243.0 ± 5.0	0.91 ± 0.04
0.1 moles	261.0 ± 5.0	221.0 ± 5.0	1.18 ± 0.06
0.5 moles	213.0 ± 5.0	171.0 ± 5.0	3.27 ± 0.02
1.0 moles	166.0 ± 5.0	136.0 ± 5.0	7.82 ± 0.07
RPMI:	261.0 ± 5.0	224.0 ± 5.0	1.84 ± 0.06
Serum-Free			
Cell Culture Media (20% FBS with 1% penn-strep)	221.0 ± 5.0	184.0 ± 5.0	1.78 ± 0.05

**Table 2 bioengineering-09-00540-t002:** Model Parameters for the Temperature and Time Dependence of DSC-Measured Heat Release.

Aqueous Solution	Temperature Dependence Parameters		Time Dependence Parameters	
	A	Tph (K)	Constant, k	Time Exponent, n
1 × PBS	0.53	272.62	3.3	1.5
10 × PBS	0.53	267.85	3.3	1.5
1 M Glycerol in 1 × PBS	0.53	270.75	2.3	1.5

## Data Availability

The data presented in this study are available on request from the corresponding author.

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
