# Peer review of "Freezing of Solute-Laden Aqueous Solutions: Kinetics of Crystallization and Heat- and Mass-Transfer-Limited Model"

_bioengineering, 2022, doi:10.3390/bioengineering9100540_

Round 1

Reviewer 1 Report

This manuscript describes the kinetics of ice crystallization obtained in biological media and the numerical model to describe the heat and mass transfer diffusion problem during freezing of a salt solution in a small cylindrical container.  This study is worthy of publication from a viewpoint of fundamental data of bioengineering.  However, the discussion on the understanding for solidification kinetics is not enough.  The referee has the following comments.

(1) Avrami constant and time exponent

The greater part of this manuscript is devoted to explain the experimental and calculation results.  The author should mention the consistency of the ice crystallization model based on the Avrami model of kinetics and the heat and mass transfer model of freezing of a salt solution in a small container.  In particular, the affinity of the physicochemical interpretation of the Avrami constant k and the Arami time exponent n to the kinetic model should be discussed.  The validity of the application of the pre-nucleated solutions at time t = 0 or 'site-saturated' conditions to this present systems should also be discussed.

 (2) Temperature and time dependence of latent heat release (Eq. 1)

The author assumed that the temperature and time dependence of latent heat release are independent of one another.  The validity of this assumption should be discussed in terms of nucleation and growth of ice with biological media in a small cylindrical container.

 (3) Characteristic of the nucleation type and the growth geometry (Table 2)

The author should mention the phenomenological influence of glycerol on Avrami constant k with the evidence based on experimental data.

 (4) Viscosity model

According to page 9 line 257, simple viscosity model was used.  The readers want to know the author's perspective on the effects of solution concentration on the solution viscosity.

Author Response

We thank the reviewer for his/her comments.  We have made several changes to the manuscript in response to his/her review as shown below: 

  1. Please, see updated footnote #2 as well as lines 193-209.
  2. Please, see lines 147-148. 
  3. Please, see lines 449-454. 
  4. The reviewer is absolutely correct, other models of viscosity exist and could have been utilized in the study.  This has now been stated in the manuscript -- lines 287-290.

Author Response

We thank the reviewer for his/her comments.  We completely agree with the reviewer that the Avrami model is clearly unsuitable for use in an non-isothermal scenario (which is the case in the present study). However, the aim of the manuscript was to assess if a simple Avrami model can capture the experimental trends such that this simple model can be used for modeling purposes.  The assumption in Eqn. (1) is purely to simplify the model and is akin to solving heat conduction problems by the principle of super-positioning.  The text has been revised to make this limitation of the model explicit and clear -- see footnote #2, lines 426-437.  

Additional changes:

  1. Please, see lines 67-77.
  2. The reviewer is correct -- if one knows a priori the amount of dissolved solids, it should be possible to estimate the excepted decrease in latent heat release during freezing; albeit with the caveat that this estimate could be significantly impacted by the presence of unfreezable or bound water.  See for example the earlier study [1] that showed that different solutions with the same amount of dissolved solids can exhibit different latent heat values.
  3. We apologize for not making this clear. Please, see lines 123-132. 
  4. The sentence has been modified.

We fully agree with the reviewer that given enough variable any model can be "fit" to describe any data and as such this "fit" is not imply understanding or knowledge of the process.  This has now been explicitly stated in the text.  Also as stated in the revised text (lines 426-437) the advantage of the Avrami model is it's simplicity which will allow simplify the computation of freezing process in tissues -- for an example of a "complicated" full model of biological freezing in tissues, please see Devireddy et al. [66].  We hope the reviewer sees "some value" in the Avrami model presented and is amenable to accepting this paper.  If not, we fully understand! 

Reviewer 3 Report

The paper concerns the determination of the latent heat of fusion in solute laden aqueous solutions. Examples are frozen food or biological samples. Experimental data are obtained using differential scanning calorimetry. Kinetic parameters are fitted to those data. A diffussion model for the freezing of a salt solution is developed and compared to experimental findings.

In general, I support publication of the work, with a few suggestions for improvement:

* How is the heat of fusion calculated in the model? Is it obtained from the heat flux -D_{T}\partial T/\partial z at the phase boundary? A formula would be helpful.

* In eq. (6), I do not understand the meaning of the parameter m.

* The cooling rate should be given in K/min instead of °C/min

* line 214: where *i=1* denotes the frozen region ...

* I found it a bit confusing that square brackets are used for literature references as well as references to equations. Maybe use ordinary brackets for equation references.

Author Response

We thank the reviewer for his/her comments. 

  1. We apologize for not making this clear.  The magnitude of latent heat fusion cannot be calculated from the model. This has now been explicitly stated in the text on lines 366-369.
  2. We apologize for not making this clear.  Please, see lines 244-247.
  3. Done. 
  4. Thank you, the typo has been fixed. 
  5. Done.

Round 2

Reviewer 1 Report

The referee confirmed the revision.

Author Response

Thank you for your review.

Reviewer 2 Report

I completely agree with the authors that the Avrami model is a good tool to describe their processes but not in the way as it is done in the present paper. As the authors agreed upon, the equations employed are not valid for non-isothermal processes, by this reason, I continue to recommend not to accept the paper in the present form.

Author Response

We have revised the manuscript to eliminate direct references to the Avrami model to refer to the model in a generic sense as an Avrami-like model of crystallization.  The isothermal assumption in the manuscript is clearly a first order approximation to traditional Avrami models and has been stated as such in the revised manuscript.